# Multiwalled Carbon Nanotubes and the Electrocatalytic Activity of *Gluconobacter oxydans* as the Basis of a Biosensor

**DOI:** 10.3390/bios9040137

**Published:** 2019-11-14

**Authors:** Yulia Plekhanova, Sergei Tarasov, Aleksandr Bykov, Natalia Prisyazhnaya, Vladimir Kolesov, Vladimir Sigaev, Maria Assunta Signore, Anatoly Reshetilov

**Affiliations:** 1FSBIS G.K. Skryabin Institute of Biochemistry and Physiology of Microorganisms, Russian Academy of Sciences, Pushchino 142290, Russia; setar25@gmail.com (S.T.); agbykov@rambler.ru (A.B.); old_copper_kettle@mail.ru (N.P.); anatol@ibpm.pushchino.ru (A.R.); 2FSBIS V.A. Kotelnikov Institute of Radio Engineering and Electronics, Russian Academy of Sciences, Moscow 125009, Russia; kvv@cplire.ru; 3FSBIS TBP RC, Branch of the FSBIS SRC Institute of Immunology, Federal Immunobiological Agency, Serpukhov 142253, Russia; vsigaev@list.ru; 4CNR, Institute for Microelectronics and Microsystems, Via Monteroni, 73100 Lecce, Italy; mariaassunta.signore@cnr.it

**Keywords:** microbial electrochemical biosensors, *Gluconobacter oxydans*, multiwalled carbon nanotubes, impedance spectroscopy, oxygen Clark-type electrode, amperometric detection

## Abstract

This paper considers the effect of multiwalled carbon nanotubes (MWCNTs) on the parameters of *Gluconobacter oxydans* microbial biosensors. MWCNTs were shown not to affect the structural integrity of microbial cells and their respiratory activity. The positive results from using MWCNTs were due to a decrease in the impedance of the electrode. The total impedance of the system decreased significantly, from 9000 kOhm (*G. oxydans*/chitosan composite) to 600 kOhm (*G. oxydans*/MWCNTs/chitosan). Modification of the amperometric biosensor with nanotubes led to an increase in the maximal signal from 65 to 869 nA for glucose and from 181 to 1048 nA for ethanol. The biosensor sensitivity also increased 4- and 5-fold, respectively, for each of the substrates. However, the addition of MWCNTs reduced the affinity of respiratory chain enzymes to their substrates (both sugars and alcohols). Moreover, the minimal detection limits were not reduced despite a sensitivity increase. The use of MWCNTs thus improved only some microbial biosensor parameters.

## 1. Introduction

Carbon nanotubes have become the nanomaterial most frequently used to modify electrochemical devices in the past decade [1,2]. Numerous papers have focused on the interaction of nanotubes with biological objects [3]. Some works show the toxic effect of carbon nanotubes on living cells, including bacterial ones [4]. At the same time, the use of bacterial cell–carbon nanotube composites in microbial biosensors and fuel cells is observed to improve their electrochemical properties [5]. The number of studies that consider the properties of systems of microbial cells coupled with nanomaterials is much lower than that of nanomaterial–enzyme studies [6,7]. A detailed investigation of this interaction is, however, of extreme importance. Such microbial biosensors can find application in the monitoring of natural and industrial water reservoirs [8] and biotechnological processes [9,10]. Great attention is given by investigators to the development of systems for biochemical oxygen demand assessment in wastewater, both communal and industrial [11]. In this method of analysis, special attention is paid to the use of novel types of microorganisms and the search for ways of their immobilization into polymers [12] with the use of nanomaterials [13]. Important results have been obtained earlier in a series of works on microbial biosensors based on strains capable of degrading surfactants [14]. Microbial biosensors are efficiently used for assessing the quality of soils [15] and foodstuffs, and for medical purposes [16]. Such works form the basis for the use of microbial biosensors in the near future.

*Gluconobacter* bacteria are often used in constructing electrodes of biosensors and microbial fuel cells [17,18]. These Gram-negative bacteria possess pyrroloquinoline quinone-dependent dehydrogenases [19] providing for electron transfer to the conducting surface of the electrode or nanomaterial. The reaction centres of these enzymes are oriented into the periplasmic space, which enables the catalysis of the biochemical reactions without the obligatory transport of substrate into the cell [20].

This study used *Gluconobacter* bacteria immobilized into a matrix from chitosan and multiwalled carbon nanotubes (MWCNTs). It was important to elucidate what effect, positive or negative, the immobilization of *Gluconobacter* bacteria into a chitosan/MWCNT matrix had on the electrocatalytic activity of microbial cells. Combining MWCNTs and chitosan could presumably lead to an increase in the number of bacteria–electrode efficient coupling sites, which facilitates electron transfer. The present study is devoted to the search for new solutions as applied to microbial biosensors.

## 2. Materials and Methods 

### 2.1. Reagents

We used sodium chloride, dipotassium hydrogen phosphate trihydrate, sodium hydroxide, glucose and acetic acid (Diakon, Russia); sodium carbonate (Khimmed, Russia); chitosan (low molecular weight), 2,6-dichlorophenolindophenol (DCPIP) sodium salt (Sigma-Aldrich, St. Louis, MO, USA); Taunit multiwalled carbon nanotubes (OD = 20–50 nm, ID = 10–20 nm, L = 0.1–2 μm) (NanoTechCentre LLC, Tambov, Russia).

Screen-printed three-contact electrodes (SPE) (Color Electronics, Russia) were used for voltammetric, amperometric, and impedance measurements. The working and counter-electrodes were made of Electrodag 6017SS graphite paste (Henkel, Germany), while the reference electrode was Ag/AgCl. The surface area of the working electrode was 7 mm^2^. 

Strain *Gluconobacter oxydans* sbsp. *industrius* VKM-1280 (All-Russian Collection of Microorganisms) was used. Cells were cultivated as described in [21].

### 2.2. Formation of Biocomposites

A biocatalyst, a 1:1 *v*/*v* mixture of a bacterial cell suspension (0.2 mg wet weight/μL) and chitosan (2% solution in 1% acetic acid [22]) was used. 

To introduce the nanomaterial, 30 μL of an MWCNT suspension (10 mg/mL) were mixed with a 20-μL suspension of bacterial cells (0.5 mg wet weight/μL). Then, 50 μL of chitosan solution were added with constant stirring. The contents of cells in both composites were equal and amounted to 0.1 mg/μL.

### 2.3. Formation of Biosensors

A schematic diagram of the microbial biosensors is shown in Figure 1. The composite in the amount of 5 μL was applied onto a fragment of Whatman GF/A (glass microfibre paper, UK), 3 × 3 mm^2^ in size, which was used as a receptor element for a Clark-type oxygen electrode. The receptor element was dried for 15–25 min at room temperature and fixed on the surface of a Clark-type oxygen electrode. The measurements were carried out in a phosphate buffer (25 mM, pH 6.5) in a 2-mL electrochemical cell at constant stirring. The measurements were carried out on an IPC-2L potentiostat (Kronas, Russia).

Composite in the amount of 5 μL was deposited onto a carbon screen-printed electrode surface and left to dry at ambient temperature. 

The voltammetric, amperometric and impedance measurements were carried out by a VersaSTAT 4 potentiostat galvanostat (Ametek Inc., Berwyn, PA, USA) or EmStat 3 potentiostat galvanostat (PalmSens, Houten, Netherlands) in a phosphate buffer (25 mM, pH 6.5) in a 1-mL electrochemical cell with constant stirring with the addition of 10 mM NaCl and a redox mediator (140 μM DCPIP). The scanning rate of 40 mV/s was used for voltammetric measurements. A 0-mV constant potential (frequency range, 40 kHz–0.02 Hz) and a voltage modulation of 10 mV were used to obtain the impedance spectra. The correct equivalent circuit for every system was picked using ZSimpWin software (EChem Software, Warminster, PA, USA). The circuit was considered correct if the fit errors of the parameters were below 10%. All chronoamperometric measurements were carried out at an applied potential of 200 mV vs. the Ag/AgCl electrode. The results of all measurements were represented as a mean value of three independent measurements. 

A JSM-6510LV 40 scanning electron microscope (JEOL, Tokyo, Japan) was used. For scanning electron microscopy (SEM), cells were washed and resuspended in sterile distilled water.

## 3. Results and Discussion

### 3.1. Scanning Electron Microscopy (SEM) and Matrix-Assisted Laser Desorption Ionization–Time of Flight Mass Spectroscopy (MALDI–TOF MS)

The conducting matrix containing bacterial cells and MWCNTs was visualized by scanning electron microscopy. Figure 2 presents micrographs of *G. oxydans* bacterial cells in the presence of MWCNTs, and chitosan gel with MWCNTs. The use of chitosan emphasizes the electrode surface structure and preserves a high electrode area-to-volume ratio.

It is seen on SEM micrographs that bacterial cells preserve their shape when immobilized in a chitosan–MWCNT matrix. It appeared important to assess whether the structure of the main enzyme complexes of bacterial cells was preserved at the impact of MWCNTs. This was determined by MALDI–TOF mass spectrometry. The spectra of bacterial cells obtained by this method are shown in Figure 3. The peaks characterized by the values of *m*/*z* (the ratio of mass *m* to charge *z* in ionized state) and the levels of intensity in relative units make it possible to identify the microorganism used in experiments as *Gluconobacter oxydans* sbsp. *industrius*. The presence of such peaks on all the spectra enables a conclusion about the preservation of the major part of *G. oxydans* protein complexes of the enzyme activity of this biocatalyst in all of the investigated composites.

### 3.2. Respiratory Activity of G. oxydans

The greatest number of microbial biosensors described in the literature are based on the change of microbial respiratory activity under the impact of external factors. As shown in Section 3.1, the shape of bacteria and the structure of their protein complexes are not damaged under the action of MWCNTs. It appeared important to assess the changes in the properties of the microbial biosensor based on a Clark-type oxygen electrode under the impact of MWCNTs. We have shown earlier [23] that the respiratory activity of *G. oxydans* cells does not change at the adsorption contact with carboxylated multiwalled carbon nanotubes. In this work, we showed that non-functionalized MWCNTs had no significant impact on the respiratory activity of *G. oxydans* cells, either (Figure 4). Thus, the use of MWCNTs as a constituent of bioreceptor elements in microbial biosensors based on a Clark-type oxygen electrode did not lead to any positive effects.

### 3.3. Impedance Spectra

It is known that MWCNTs possess unique physical properties, such as high electrical conductivity and a large surface area [24]. For this reason, their use can be promising in electrochemical microbial biosensors for increasing the active surface area of the electrode and reducing its impedance. To study the electrochemical properties of the chosen composites, we used microbial biosensors based on screen-printed carbon electrodes. The total impedance of the bioelectrochemical systems based on the chosen composites is given in the Nyquist diagrams in Figure 5. We observed a significant decrease in both the total impedance of the system (from 9000 kOhm to 600 kOhm) and the charge transfer resistance of the system in the transition from the *G. oxydans*/chitosan composite to *G. oxydans*/MWCNTs/chitosan. A ~15-fold decrease in the total impedance was due to an extremely low charge transfer resistance of MWCNTs, described in the literature on numerous occasions [25]. However, it should be noted that chitosan is a poorly conducting polymer [26] and its use deteriorates the general electrochemical properties of the biosensor. Furthermore, the total impedance of the *G. oxydans*/MWCNTs/chitosan composite is significantly affected by the capacitive component. Carbon nanotubes level off the negative effect of chitosan and accelerate the total rate of charge transfer from microorganisms to the electrode surface. Hence, it is expedient to use MWCNTs in second- and third-generation amperometric biosensors.

### 3.4. Amperometric Biosensors Based on G. oxydans/Chitosan and G. oxydans/MWCNTs/Chitosan Biocomposites

The presence of MWCNTs in a biosensor’s composite leads to a decrease in the electrode’s impedance. Correspondingly, the addition of MWCNTs to the composite applied to the surface of a screen-printed electrode results in the increase of anodic and cathodic peak currents on cyclic voltammograms. The increase in anodic currents depends on the content of MWCNTs in the composite (Figure 6). A MWCNTs concentration of 2.1 μg/mm^2^ was used in further experiments to achieve the best possible electrochemical properties of the microbial biosensors.

The effects of MWCNTs on the analytical characteristics of amperometric microbial biosensor were considered for two types of substrates: glucose (sugars) and ethanol (alcohols). The measured biosensor signal was the change in the electrode current after the addition of a substrate. The corresponding calibration curves are given in Figure 7 for glucose and ethanol.

The analytical parameters of biosensors for glucose and ethanol assays based on both composites are given in Table 1.

The smaller the value of *K*_M_, the higher the affinity of the enzyme to the substrate is [27]. As seen from the data obtained, a modification of the polymer by MWCNTs leads to an increase in the apparent Michaelis constants, i.e., the affinity of bacterial cells’ enzymes to the substrate (both glucose and ethanol) decreases. In the case of glucose dehydrogenase, the affinity to the substrate decreases 3.5-fold; in the case of the alcohol dehydrogenase it is only 1.3-fold, which is indicative of a stronger effect of MWCNTs on the glucose dehydrogenase. At the same time, with the use of a *G. oxydans*/MWCNTs/chitosan composite the biosensor signal is observed to be significantly increased. The value of the maximum rates of reaction in response to the introduction of substrates for modified biosensors is increased 13.5-fold in the case of glucose and 5.8-fold with the introduction of ethanol. 

The Hill coefficient is a dimensionless value that characterizes the cooperativity of ligand binding by the enzyme or receptor. A positive cooperativity is characterized by the fact that at the attachment of the ligand to an active centre of the enzyme the attachment of the subsequent ligands to the other active centres is facilitated [27]. As seen from the data obtained, in all cases the Hill coefficient is greater than 1, i.e., a positive cooperativity of the enzymes is observed for the bacterial alcohol and glucose dehydrogenases. The use of a *G. oxydans*/MWCNT/chitosan composite leads to an increase in the value of this coefficient, i.e., an increase in the cooperativity of the enzymes. 

In [28], the apparent Michaelis constant for alcohol dehydrogenase as a component of a MWCNTs/chitosan/alcohol dehydrogenase composite has been shown to be 0.38 mM; the detection limit is 0.52 μM and the biosensor sensitivity is 0.1646 A/M·cm^2^. Our work showed (see Table 1) that the value of the apparent Michaelis constant for alcohol dehydrogenase incorporated into cells as a *G. oxydans*/MWCNTs/chitosan composite constituent was close to that of the free enzyme (0.315 mM); the minimal detection limit was lower (0.015 mM), as well as the sensitivity (by approximately an order of magnitude), which is normal for microbial biosensors as compared with enzymatic biosensors.

## 4. Conclusions

This work aimed to study microbial biosensors based on combinations of MWCNTs, chitosan and *G. oxydans* cells. We showed the effect of an MWCNTs/chitosan composite on the structure and integrity of protein complexes, as well as on the respiratory activity of bacteria and their electrochemical parameters. The impact of MWCNTs did not inhibit the activity of microorganisms and did not disturb the integrity of their cell envelopes. At the same time, the effect of MWCNTs as a component of a *G. oxydans*/MWCNTs/chitosan composite was not limited by a decrease in the charge transfer resistance. The use of this composite as a part of an amperometric microbial biosensor was shown to reduce the affinity of respiratory chain enzymes to their substrates (both sugars and alcohols) against the background of the increase in the general sensitivity of the electrochemical biosensor. Thus, it can be assumed that MWCNTs mainly affect the reaction centres of the enzymes, without disturbing the integrity of enzyme complexes and the structure of cells. The obtained results can be used in the development of modern microbial biosensors and microbial fuel cells based on carbon nanomaterials.

## Figures and Tables

**Figure 1 biosensors-09-00137-f001:**
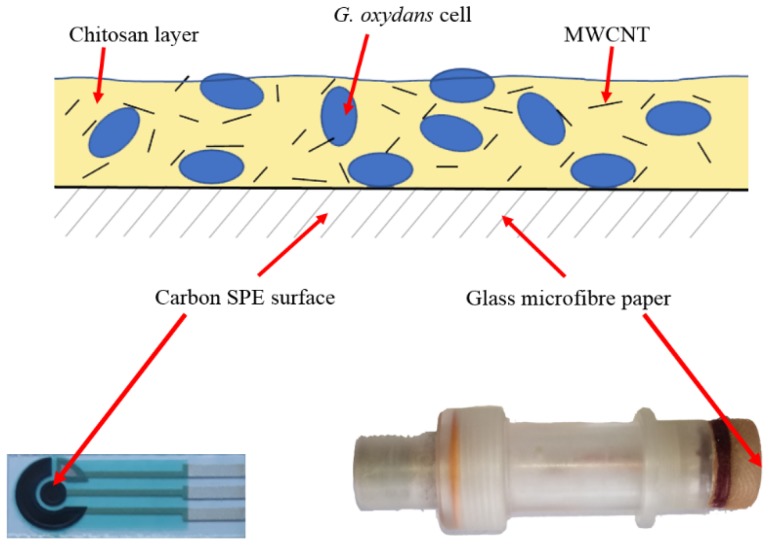
Schematic diagram of microbial biosensors based on a carbon screen-printed electrode and an oxygen electrode.

**Figure 2 biosensors-09-00137-f002:**
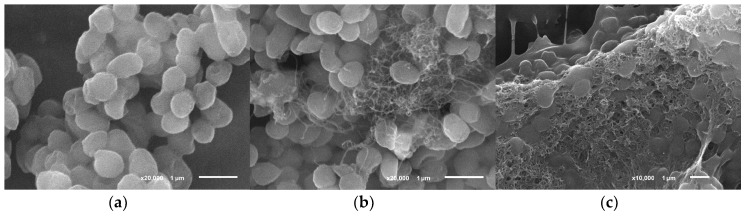
Scanning electron micrographs of the surface of a graphite electrode covered with *G. oxydans* cells (**a**); *G. oxydans* cells with MWCNTs (**b**); *G. oxydans* cells in chitosan gel with MWCNTs (**c**).

**Figure 3 biosensors-09-00137-f003:**
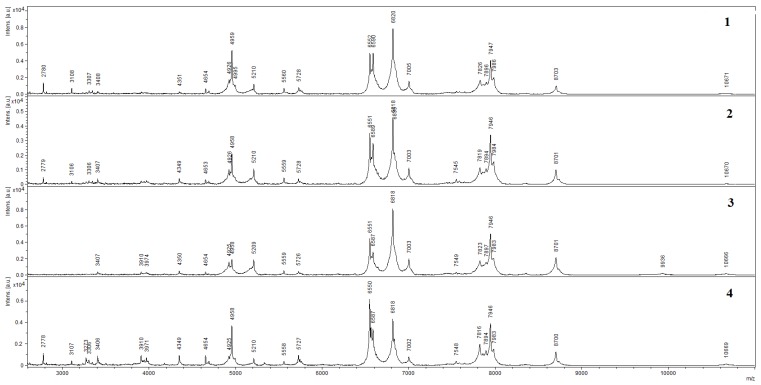
MALDI–TOF mass spectra: *G. oxydans* (**1**); *G. oxydans*/MWCNTs/chitosan (**2**); *G. oxydans*/chitosan (**3**); *G. oxydans*/MWCNTs (**4**).

**Figure 4 biosensors-09-00137-f004:**
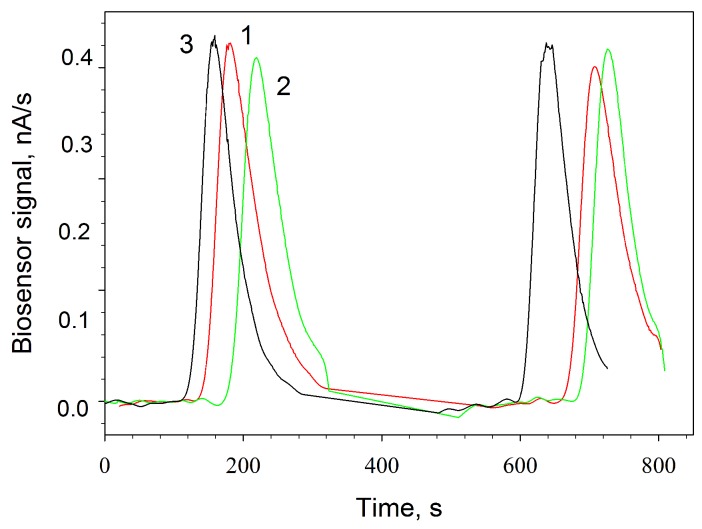
Signals of a biosensor based on a Clark-type oxygen electrode with a bioreceptor from *G. oxydans* (1), *G. oxydans*/chitosan (2), *G. oxydans*/MWCNTs/chitosan (3) in response to the addition of 1-mM ethanol.

**Figure 5 biosensors-09-00137-f005:**
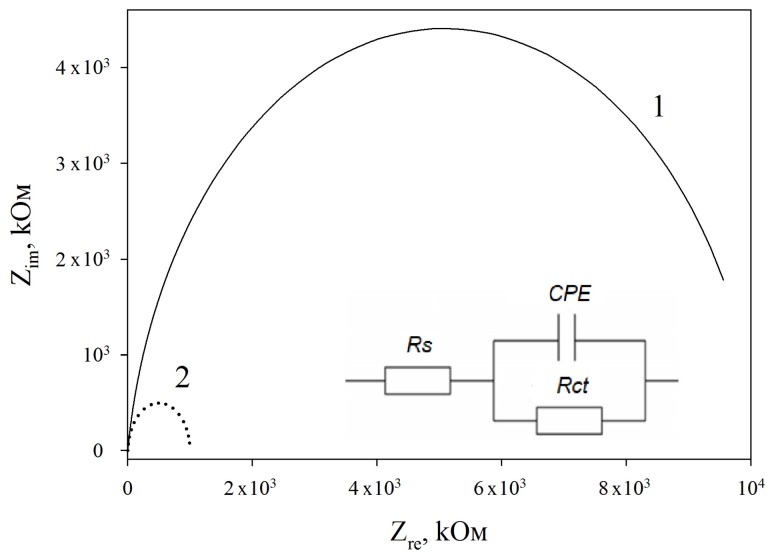
Nyquist diagrams for a biosensor based on *G. oxydans*/chitosan (1) and *G. oxydans*/MWCNT/chitosan (2) composites in the presence of 140 μM DCPIP. The spectra were modelled with the electrical equivalent circuit shown in the inset.

**Figure 6 biosensors-09-00137-f006:**
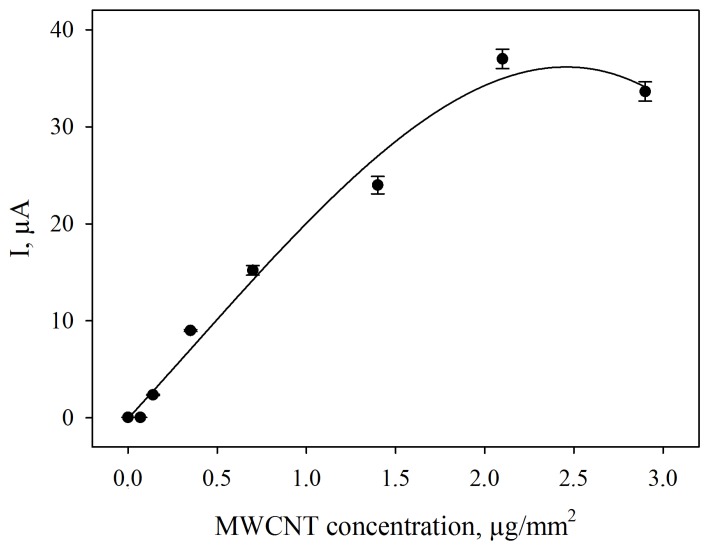
Effect of MWCNT concentration on anodic currents on the cyclic voltammograms in the presence of 1-mM ethanol in a 0.1-M phosphate buffer solution containing 140 μM DCPIP at a *G. oxydans*/MWCNTs/chitosan composite-modified carbon screen-printed electrode. Scan rate: 40 mVs^−1^.

**Figure 7 biosensors-09-00137-f007:**
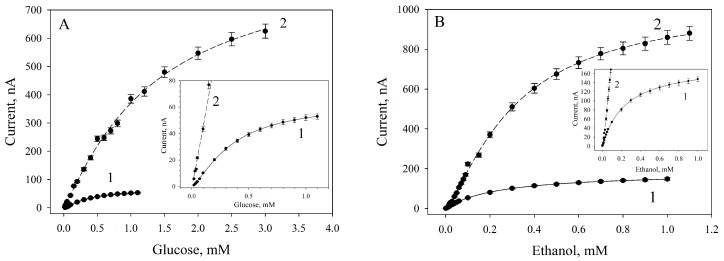
Calibration curves for glucose (**A**) and ethanol (**B**) analysis with the values averaged from three consecutive calibrations of a biosensor based on *G. oxydans*/chitosan (1) and *G. oxydans*/MWCNT/chitosan (2) composites. Insets show zoomed-in sections of low-current regions of the graphs.

**Table 1 biosensors-09-00137-t001:** Analytical characteristics of microbial biosensors for glucose and ethanol assays.

Substrate		Glucose		Ethanol	
	Modification	*G. oxydans*/Chitosan	*G. oxydans*/MWCNTs/Chitosan	*G. oxydans*/Chitosan	*G. oxydans*/MWCNTs/Chitosan
Parameter					
*V*_max_, nA		64.55 ± 1.99	869.26 ± 76.02	181.36 ± 2.65	1048.31 ± 41.88
*K*_m_, мM		0.36 ± 0.02	1.27 ± 0.22	0.24 ± 0.01	0.32 ± 0.03
*h*		1.36 ± 0.06	1.14 ± 0.09	1.02 ±0.02	1.30 ± 0.07
Linear range of detection, mM		0.10–0.60	0.17–1.82	0.07–0.34	0.06–0.50
Regression equation for the linear segment		*y* = 65.80*x* + 6.89	*y* = 259.06*x* + 92.18	*y* = 240.94*x* + 30.57	*y* = 1257.50*x* + 99.49
Correlation coefficient, *R*^2^		0.97	0.97	0.97	0.97
Sensitivity coefficient, μA/mM		65.80	259.06	240.94	1257.5
Minimal detection limit, mM		0.04	0.04	0.003	0.015
Detection range, mM		0.04–1.00	0.04–2.50	0.003–0.700	0.015–1.000

Note: An equation describing the calibration dependences: $V=\frac{V_{max}S^{h}}{K_{M}^{h}+\;S^{h}}$. The correlation coefficient for calibration dependences and for the regression equation for the linear segment of *R*^2^ is 0.99. Mean values from three measurements and standard deviations from the mean values are given.

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
