# Peer review of "Multiwalled Carbon Nanotubes and the Electrocatalytic Activity of Gluconobacter oxydans as the Basis of a Biosensor"

_biosensors, 2019, doi:10.3390/bios9040137_

Round 1
Reviewer 1 Report
Dear Author,
I think that your work describes an interesting activity, but it needs to improve before publication. Please consider the following points for redrafting:
-) your draft needs a mild English form revision;
-) r. 20: please define the acronym PQQ before using it;
-) rr. 31-48: this is suitable for a Review Article, as it describes the advances in the field starting from the very origins; please remove it or focus it on modern achievements;
-) r. 73: please clarify your electrochemical set up. Did you use a carbon worker with both reference and counter made also in carbon? What are the dimensions of your worker? Did you manage to drop your functionalization solution volume right on the electrode or do you use a mask?
-) rr. 92-93: why do you use two different electrochemical instruments?
-) r. 94: please clarify the role of DCPIP in your detection strategy;
-) r. 97: please indicate the fit parameters values and the fit error of your models;
-) Fig. 1: the image in a) and b) look too similar;
-) r. 143: please correct “spectras” with “spectra”;
-) Fig. 4: the spectra are clearly related to a parallel RC, but have you tried with a redox mediator to investigate the diffusive component?
-) Fig. 5: what is the numerosity of your experiments, i.e., how many different sensors did you test to obtain the curve?
-) Fig. 6A: please explain why the maximum concentration of 1 is different from the one of 2;
-) Fig. 6A and B: please substitute “Biosensor signal” with the proper measured parameter, e.g., anodic peak current or similar;
-) Fig. 6A and B: please show the error bars;
-) Fig. 6B: the “Biosensor signal” obtained for 1 mM ethanol in curve 2 (0.8 uA) is different from that obtained during the calibration of Fig. 5 (about 40 uA). Please give an explanation.
Author Response
Point 1: your draft needs a mild English form revision;
Response 1: The English of the manuscript was revised where appropriate.
Point 2: r. 20: please define the acronym PQQ before using it;
Response 2: The acronym was replaced by its full form (pyrroloquinoline quinone)
Point 3: rr. 31-48: this is suitable for a Review Article, as it describes the advances in the field starting from the very origins; please remove it or focus it on modern achievements;
Response 3: We changed Introduction to include some modern achievements (lines 33-55).
Point 4: r. 73: please clarify your electrochemical set up. Did you use a carbon worker with both reference and counter made also in carbon? What are the dimensions of your worker? Did you manage to drop your functionalization solution volume right on the electrode or do you use a mask?
Response 4: We added clarification for our setup in lines 69-72. We used carbon working and counter electrodes and Ag/AgCl reference electrode. The surface area of working electrode was 7 mm2. We dropped our solutions right on the electrode surface without additional masks.
Point 5: rr. 92-93: why do you use two different electrochemical instruments?
Response 5 We used VersaSTAT 4 for impedance measurements mostly and EmStat 3 for other experiments. Also, we added a second instrument to evaluate the reproducibility of our results between two different devices.
Point 6: r. 94: please clarify the role of DCPIP in your detection strategy;
Response 6: we added the clarification in line 99. DCPIP was used as a redox mediator.
Point 7: r. 97: please indicate the fit parameters values and the fit error of your models;
Response 7: The circuit was considered correct if the fit errors of the parameters were below 10% (lines 103-104)
Point 8: Fig. 1: the image in a) and b) look too similar;
Response 8: It was a mistake on our part; we replaced Fig 1 b) (now Fig. 2 b)
Point 9: r. 143: please correct “spectras” with “spectra”;
Response 9: Done
Point 10: Fig. 4: the spectra are clearly related to a parallel RC, but have you tried with a redox mediator to investigate the diffusive component?
Response 10: We added the clarification in the legend to Figure 5 (previously Fig 4). All our impedance spectra were obtained in the presence of redox mediator. There are no 45-degree slopes on our spectra which could correspond to Warburg element so there are no diffusive limitation in our systems.
Point 11 Fig. 5: what is the numerosity of your experiments, i.e., how many different sensors did you test to obtain the curve?
Response 11: Each point measured for each graph in this work was repeated three times and no significant deviations have been found. Furthermore, the measurements were repeated by two other electrodes, which were fabricated at different times and gave almost the same response under the same conditions. We added clarifications in Materials and Methods (lines 105-106)
Point 12: Fig. 6A: please explain why the maximum concentration of 1 is different from the one of 2;
Response 12: Detection range of G. oxydans /MWCNTs /chitosan biosensor is greater than the G. oxydans /chitosan one. We finished the calibration curve when the current signal for the next substrate concentration did not differ from the current signal for the previous substrate concentration.
Point 13: Fig. 6A and B: please substitute “Biosensor signal” with the proper measured parameter, e.g., anodic peak current or similar; Fig. 6A and B: please show the error bars;
Response 13: We substituted “Biosensor signal” with the measured parameter (current) as these responses are derived from current-time measurements. Also, we added error bars and insets to represent the data more clearly.
Point 14: Fig. 6B: the “Biosensor signal” obtained for 1 mM ethanol in curve 2 (0.8 uA) is different from that obtained during the calibration of Fig. 5 (about 40 uA). Please give an explanation.
Response 14: Figure 6 shows the calibration dependence obtained from the anode currents on cyclic voltammograms taken at a rate of 40 mV/s. In Fig. 7B (previously Fig. 6B), we showed the calibration curves obtained for amperometric biosensors, i.e. biosensor signal under an applied potential of 200 mV. Since these are different measurement methods, the signals along the Y axis are different.

Reviewer 2 Report
The critical comments are as follows:
1. The whole abstract is full of generality and needs rewrite as it is qualitatively descriptive in its current form. It doesn’t deduce the justification of the results obtained. Normally an abstract should state briefly the purpose of the study undertaken and meaningful conclusions based on the obtained results. Hence, this needs rewriting. I would expect brief, yet concise, the quantitative data description of the results in the abstract.
2. Throughout the manuscript, the level of English used is not up to the standard of the journal. The sentences are long and badly worded with repetitive words. Please consider breaking longer sentences into smaller fragments for easy understanding. Authors are advised to seek help from a native English speaker. For example, L38-40.
3. In methodology, the inclusion of a scheme for the development of biosensor is recommended.
4. Mostly the Figures are poor in quality and lines are overlapping making it difficult to read. Reconstruct.
5. Tables: What was the sample size? It needs to be clearly mentioned. Add a footnote explaining the coated values were taken from the duplicate/triplicate samples.
6. Editorial issues: The Latin names and Greek letters should be presented in italic in the whole manuscript, the unit presentation should be unified in the whole manuscript, abbreviations presentation should be unified.
7. Referencing is not right and consistent. Most of the reference are as old as 90s. This should be improved as there are many reports available from the year 2018-2019. Enrich the reference list with recent literature.
Author Response
Point 1. The whole abstract is full of generality and needs rewrite as it is qualitatively descriptive in its current form. It doesn’t deduce the justification of the results obtained. Normally an abstract should state briefly the purpose of the study undertaken and meaningful conclusions based on the obtained results. Hence, this needs rewriting. I would expect brief, yet concise, the quantitative data description of the results in the abstract.
Response 1: We agree that our Abstract was too generic. We added more quantitative data based on obtained results (lines 19-28).
Point 2. Throughout the manuscript, the level of English used is not up to the standard of the journal. The sentences are long and badly worded with repetitive words. Please consider breaking longer sentences into smaller fragments for easy understanding. Authors are advised to seek help from a native English speaker. For example, L38-40.
Response 2: We tried to improve our English with the help of a native English speaker and broke longer sentences into smaller fragments where possible.
Point 3. In methodology, the inclusion of a scheme for the development of biosensor is recommended.
Response 3: We added a scheme for the development of both biosensors (Fig. 1.)
Point 4. Mostly the Figures are poor in quality and lines are overlapping making it difficult to read. Reconstruct.
Response 4: We changed figures 3 and 4 so now they are easier to read without overlapping. Also, we added higher quality version of Fig. 2 to our article.
Point 5. Tables: What was the sample size? It needs to be clearly mentioned. Add a footnote explaining the coated values were taken from the duplicate/triplicate samples.
Response 5: We added the clarification about the sample size to Materials and methods section (lines 105-106) and a footnote under the table.
Point 6. Editorial issues: The Latin names and Greek letters should be presented in italic in the whole manuscript, the unit presentation should be unified in the whole manuscript, abbreviations presentation should be unified.
Response 6: We presented all of Latin names and Greek letters in italic.
Point 7. Referencing is not right and consistent. Most of the reference are as old as 90s. This should be improved as there are many reports available from the year 2018-2019. Enrich the reference list with recent literature.
Response 7: We changed our Introduction section to describe some modern achievements in the field. Therefore, our reference list is updated with recent literature now.

Round 2
Reviewer 1 Report
Dear Authors,
thanks for your redrafting efforts.
Kind regards.
Reviewer 2 Report
The revised version reads well. The authors have addressed all the comments raised in the last review. This manuscript can now be accepted for publication.